# ResVR: Joint Rescaling and Viewport Rendering of Omnidirectional Images

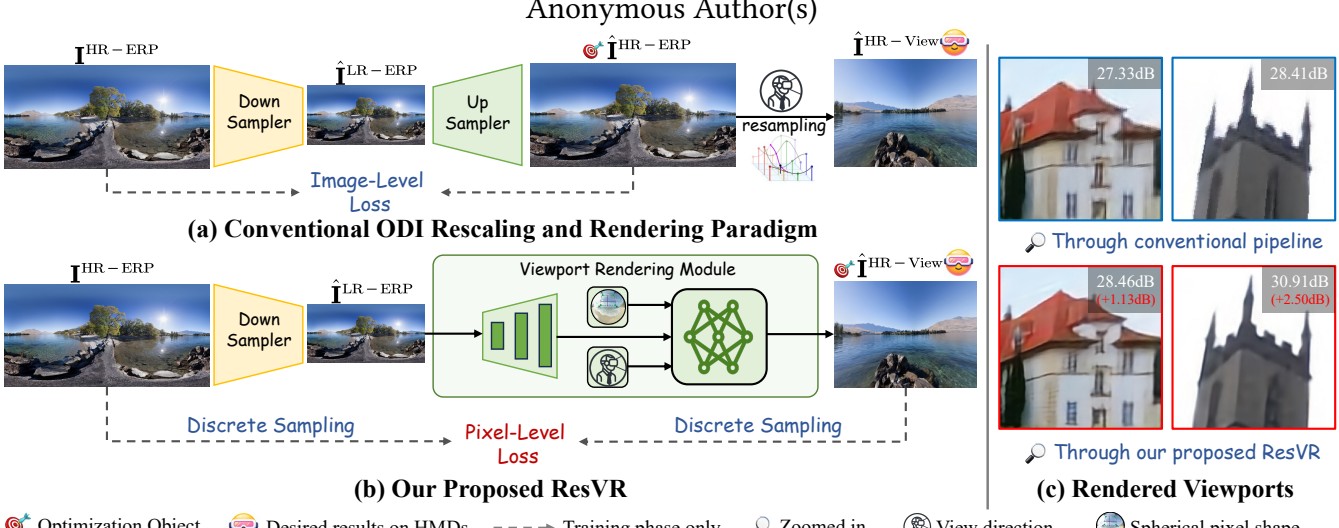

Figure 1: Proposed ResVR compared to previous ODI rescaling and viewport rendering paradigms. **(a)** Conventional methods focus on improving the quality of rescaled ERP images, resulting in inferior visual experiences. **(b)** Considering the fact that the desired content viewed on HMDs is a rendered viewport instead of an ERP image, our ResVR directly optimizes the quality of the final viewport for users through a novel discrete pixel sampling strategy and a spherical pixel shape representation technique. **(c)** Visual and PSNR comparisons of rendered viewports between the two pipelines in (a) and (b) on user HMDs.

## ABSTRACT

With the advent of virtual reality technology, omnidirectional image (ODI) rescaling techniques are increasingly embraced for reducing transmitted and stored file sizes while preserving high image quality. Despite this progress, current ODI rescaling methods predominantly focus on enhancing the quality of images in equirectangular projection (ERP) format, which overlooks the fact that the content viewed on head mounted displays (HMDs) is actually a rendered viewport instead of an ERP image. In this work, we emphasize that focusing solely on ERP quality results in inferior viewport visual experiences for users. Thus, we propose **ResVR**, which is the first comprehensive framework for the joint Rescaling and Viewport Rendering of ODIs. ResVR allows obtaining LR ERP images for transmission while rendering high-quality viewports for users to watch on HMDs. In our ResVR, a novel discrete pixel sampling strategy is developed to tackle the complex mapping between the viewport and ERP, enabling end-to-end training of ResVR pipeline. Furthermore, a spherical pixel shape representation technique is innovatively derived from spherical differentiation to significantly improve the visual quality of rendered viewports. Extensive experiments demonstrate that our ResVR outperforms existing methods in viewport rendering tasks across different fields of view, resolutions, and view directions while keeping a low transmission overhead[1].

## CCS CONCEPTS

• **Computing methodologies** → **Computer vision**.

## KEYWORDS

Omnidirectional Image, Image Rescaling, Viewport Rendering

## 1 INTRODUCTION

With the growing interest in virtual reality and augmented reality, omnidirectional images (ODIs), also referred to as 360° or panoramic images, attract great attention within the computer vision community for their immersive and interactive capabilities. Although ODIs can capture scenes across a comprehensive 360°×180° views, head-mounted displays (HMDs) often present a limited field-of-view (FoV), necessitating resolutions as high as 4K×8K [2] to preserve details in a small viewport. High-resolution (HR) ODIs are typically stored on cloud servers by platforms of virtual reality media, requiring real-time download by users. This can degrade the visual experience of users, particularly under poor internet conditions.

Image rescaling [12, 26, 46–48, 50] emerges as an effective method to reduce image file size for storage and transmission while preserving high quality in the reconstructed images at the user end. This technique first downscales HR images to low-resolution (LR) ones that keep the most important visual details, then upscales them back to their original HR versions. It not only minimizes the file size of LR images but also maintains the quality of reconstructed HR images [35], thus becoming a straightforward approach for efficient transmission and storage for ODIs from cloud servers to user HMDs. However, the prevalent storage and transmission format for ODIs, i.e., the equirectangular projection (ERP), has directed state-of-the-art ODI rescaling works [16] to focus on enhancing the quality of ERP images. As depicted in Fig. 1 (a), after receiving an LR ODI,

---

[1]The complete code and pre-trained models of our method will be made available.

the typical process on HMDs is two-step, involving (1) upscaling HR ODIs in ERP format from LR images ($\hat{\mathbf{I}}^{\text{LR-ERP}} \mapsto \hat{\mathbf{I}}^{\text{HR-ERP}}$), and (2) projecting them onto the viewport ($\hat{\mathbf{I}}^{\text{HR-ERP}} \mapsto \hat{\mathbf{I}}^{\text{HR-View}}$) using traditional interpolation methods such as bilinear. This pipeline does not fully account for the ultimate rendered image of the HMD viewport $\hat{\mathbf{I}}^{\text{HR-View}}$, particularly lacking optimization for the final viewing experience. As Fig. 1 (c) shows, the quality of images seen by the user on HMDs can be significantly lower than anticipated.

In this paper, we point out that (1) the content viewed on HMDs is actually a rendered viewport, not an ERP image, and (2) focusing solely on the quality of ERP images will result in sub-optimal viewport visual experiences. To improve the viewing experience for users, there is a need to develop a comprehensive solution that is optimized for end-to-end ODI processing from the storage of ERP images to the display of the final viewport. To this end, we propose **ResVR**, a novel framework for joint Rescaling and Viewport Rendering of ODIs, marking an innovative step towards comprehensive end-to-end ODI processing. As shown in Fig. 1 (b), ResVR aligns the optimization of network parameters with our primary goal of improving the quality of the final viewport. By utilizing such a new methodology, the viewports rendered through our ResVR framework exhibit enhanced details and fewer artifacts compared to those produced by conventional pipelines, as shown in Fig. 1 (c).

In our proposed ResVR, HR ODIs are firstly embedded into LR images to facilitate efficient transmission. Then, HR viewports are directly rendered from LR ERP images on HMDs. This process ($\hat{\mathbf{I}}^{\text{LR-ERP}} \mapsto \hat{\mathbf{I}}^{\text{HR-View}}$) does not need to produce HR ERP images. To deal with the irregular correspondence between the ERP area and the viewport, which hinders the joint optimization of downscaling and viewport rendering using traditional image-level loss training methods, we develop a discrete pixel sampling strategy. In each training iteration, this strategy is used to randomly sample paired sets of ground truth pixels and the reconstructed ones on viewports, thus making the end-to-end learning of our entire ResVR pipeline feasible in implementation. Furthermore, to enhance our ResVR's awareness of the positions of different viewport pixels on the spherical surface, we introduce a technique for spherical pixel shape representation. This technique employs spherical differentiation to calculate the geometric orientation and curvature of various viewport areas, offering positional information that contains more precise spherical attributes than existing 2D image representation methods [22]. This advanced representation effectively improves the quality of the final viewport, especially in regions of high latitude and longitude. Extensive experiments on various panoramic image datasets show that ResVR outperforms existing methods in multiple viewport rendering tasks across different FoVs, view directions, and resolutions. In summary, our contributions are:

❏ We propose ResVR, a novel framework for the comprehensive processing of omnidirectional images, seamlessly integrates image **Res**caling with **V**iewport **R**endering. Our ResVR effectively balances the transmission efficiency and users' visual experience.

❏ We develop a discrete pixel sampling strategy to tackle the complex correspondence between viewport and equirectangular projection (ERP) areas within our framework. This strategy makes the end-to-end training of our whole processing pipeline feasible.

❏ We introduce a spherical pixel shape representation technique based on spherical differentiation to guide viewport rendering, which significantly enhances the visual quality of the final viewport.

❏ Extensive empirical evaluations on various panoramic image datasets exhibit that ResVR consistently achieves new state-of-the-art visual quality while maintaining a low transmission bitrate.

## 2 RELATED WORK

### 2.1 Omnidirectional Image Super-Resolution

Image super-resolution (SR) seeks to construct HR images from LR ones. Since the advent of deep neural networks (DNNs) in SR-CNN [15], subsequent studies [10, 13, 25, 27, 28, 42, 43, 60, 61] have significantly advanced SR performance beyond traditional methods. In the specific context of omnidirectional image super-resolution (ODISR), DNN-based approaches have been tailored to account for the unique latitude-based characteristics of ODIs [1, 5–7, 14, 29, 31, 33, 39, 57, 58]. For example, LAU-Net [14] divides the entire ERP image into latitude-based patches for separate upscaling. SphereSR [57] introduces the spherical local implicit image function (SLIIF) alongside a novel feature extraction module to leverage information from arbitrary projection types. OSRT [58] employs a distortion-aware transformer targeting dimension-related distortions in ERP images. OPDN [39] introduces a dual-stage framework incorporating a position-aware deformable network. Despite these noteworthy advancements, the majority of these methodologies presuppose a fixed downscaling approach (e.g. bicubic [30]) and overlook high-frequency components from HR inputs, thus limiting the quality of reconstructed details in SR ODIs.

### 2.2 Image Rescaling

Different from SR, image rescaling focuses on downscaling HR images to create visually pleasing LR images that retain essential information for accurate HR reconstruction. Recently, invertible neural network (INN) [4, 9, 32] becomes a representative framework for image rescaling [12, 26, 34, 38, 46, 47, 51, 54], offering a direct route to inversely map the downscaled images back to HR. For instance, IRN [46, 47] is the first attempt to model image downscaling and upscaling using invertible transmissions. Liang *et al.* [26] formulate high-frequency components in INNs as a conditional distribution on the LR image. HyberThumbnail [35] employs an asymmetric encoder-decoder architecture for real-time reconstruction of 6K images and also optimizes the JPEG compression process [41, 44, 49]. Very recently, DINN [16] makes the first attempt to apply image rescaling to ODI and highlights the significance of leveraging ERP's latitude characteristics by developing a latitude-aware conditional mechanism. However, existing ODI rescaling methods focus solely on improving the quality of ERP images. In contrast, our ResVR innovatively optimizes the quality of the final viewport, offering new improvements orthogonal to previous ODI rescaling methodologies.

### 2.3 Viewport Rendering of ODIs

ODIs are designed to encapsulate a full spherical view, enabling an immersive viewing experience. However, when viewed through HMDs towards a particular direction, only a specific viewport is displayed [18, 19]. Achieving high resolution and quality in these viewports is crucial for immersive experiences, as highlighted by

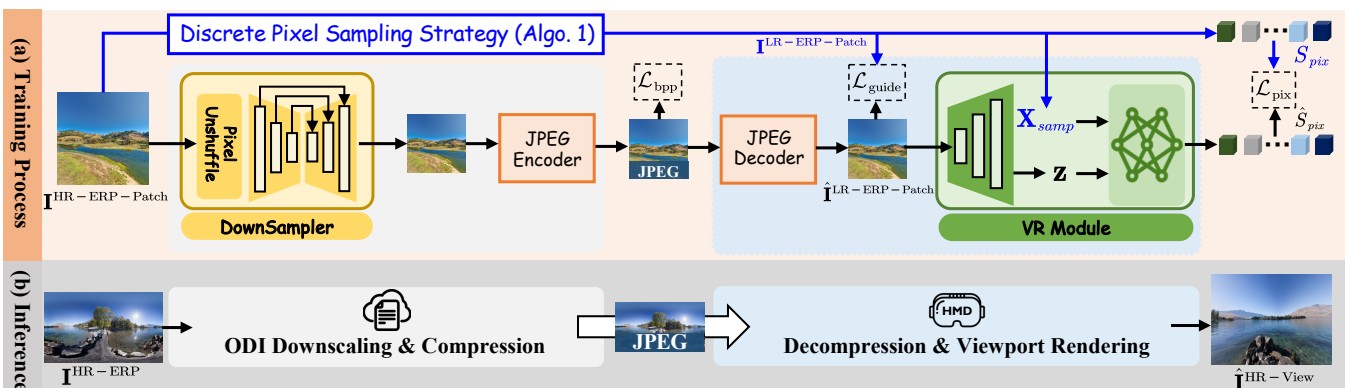

**Figure 2: Overview of our proposed ResVR framework. The comprehensive ODI processing of ResVR contains two sequential steps: (1) ODI Downscaling & Compression and (2) Decompression & Viewport Rendering. (a) In the training process, HR ERP patches $I^{HR\text{-}ERP\text{-}Patch}$ are randomly sampled through our proposed discrete pixel sampling strategy (Algo. 1) to generate the guided LR patches $I^{LR\text{-}ERP\text{-}Patch}$, query coordinates $X_{samp}$ and the set of ground truth pixels $S_{pix}$. This strategy innovatively makes the end-to-end training of ResVR feasible in implementation. (b) During inference, our trained ResVR model can be directly applied for joined rescaling and viewpoint rendering of given HR ERP images from the cloud server to user HMDs.**

various ODI visual quality assessment techniques [52, 53, 55, 56]. As a widely adopted method for viewport rendering [8], perspective projection employs a series of pixel mapping and resampling operations to effectively implement image warping. To reduce the interpolation-induced blurriness and artifacts, SRWarp [37] reinterprets image warping as an SR problem and introduces a differentiable warping module. LTEW [22] uses a continuous neural representation [11, 20, 23] for image warping by taking advantage of both Fourier features and spatially-varying Jacobian matrices. LeRF [24] assigns spatially varying steerable resampling functions to pixels, learning their orientations for continuous function prediction. Different from the above methods that focus on warping, our ResVR considers comprehensive ODI processing, serving as a new framework while enjoying high viewpoint rendering quality.

## 3 METHODOLOGY

In this section, we begin with a concise review of the viewport rendering process (Sec. 3.1). Following this, we provide an overview of our ResVR framework (Sec. 3.2), which is illustrated in Fig. 2. We then elaborate on the proposed discrete pixel sampling strategy (Sec. 3.3) and spherical pixel shape representation technique (Sec. 3.4). The training objectives are detailed in Sec. 3.5.

### 3.1 Preliminaries of Viewport Rendering

ODIs inherently provide a full spherical view. However, when viewed through HMDs directed towards a specific direction, only the corresponding viewport is displayed. This viewport appears as a 2D image, derived through perspective projection [21] from a segment of the spherical image. To formalize this process, we consider the view direction in spherical coordinates $(\theta_c, \phi_c)$, along with the horizontal and vertical fields of view $(F_h, F_v)$, and the height and width of the viewport $(h_v, w_v)$. An invertible coordinate mapping $f : X \mapsto Y$ is established, where $X := \{x | x \in \mathbb{R}^2\}$ denotes the coordinate space of ERP, and $Y := \{y | y \in \mathbb{R}^2\}$ represents the coordinate space of the viewport to be rendered. In practice, the coordinates $Y_{view}$ on the viewport are initially determined by $(h_v, w_v)$, and then

the corresponding coordinates on ERP are obtained through backward mapping $X_{view} = f^{-1}(Y_{view})$. The rendering of the viewport is achieved through resampling techniques, such as interpolation, ensuring that this process remains fully differentiable. Additional mathematical details of perspective projection and viewport rendering are elaborated in the supplementary material.

### 3.2 Overview of ResVR

An overview of proposed joint Rescaling and Viewport Rendering (**ResVR**) of ODIs is presented in Fig. 2. The ODI processing of ResVR contains two steps: (1) ODI Downscaling & Compression: An HR ERP image is firstly downsampled and compressed to an LR ERP JPEG image for efficient transmission from cloud server to user HMDs, and (2) Decompression & Viewport Rendering: The LR ERP image is then decompressed and rendered to HR viewports on HMDs through our viewport rendering (VR) module.

**ODI Downscaling & Compression:** Given an HR ERP image $I^{HR\text{-}ERP} \in \mathbb{R}^{3 \times H \times W}$, its LR representation is firstly generated through our downsampler, where $s$ is the rescaling factor. The downsampler is a U-Net [36] with dense blocks [17]. To further decrease the size of the transmitted image file, a JPEG encoder is employed for the LR representation to obtain a JPEG image. Concretely, we follow [35] to predict adaptive quantization tables for each image. Finally, the LR JPEG image is obtained, and adaptive quantization tables and quantized DCT coefficients are also encoded into the JPEG file. More details about the downsampler, the adaptive quantization table prediction module, and the training process of learned compression are provided in the supplementary material.

**Decompression & Viewport Rendering:** After receiving the LR JPEG image, $\hat{I}^{LR\text{-}ERP} \in \mathbb{R}^{3 \times \frac{H}{s} \times \frac{W}{s}}$ is firstly reconstructed by the JPEG decoder through inverse discrete cosine transformation (IDCT). Then our goal is to render high-resolution viewport $\hat{I}^{HR\text{-}View} \in \mathbb{R}^{3 \times h_v \times w_v}$ directly from $\hat{I}^{LR\text{-}ERP}$. Inspired by recent implicit neural representation methods [11, 22, 23], we develop a viewport rendering (VR) module to predict pixel values of the query coordinates

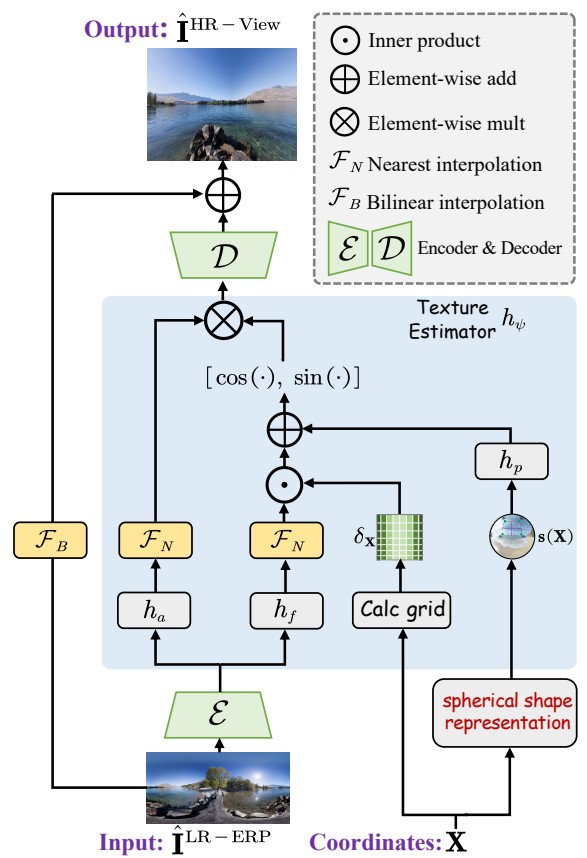

Output: $\hat{\mathbf{I}}^{HR-View}$

Input: $\hat{\mathbf{I}}^{LR-ERP}$    Coordinates: $\mathbf{X}$

Figure 3: Illustration of the VR module, which consists of an encoder $\mathcal{E}$, a local texture estimator $h_\psi$, and an MLP decoder $\mathcal{D}$. Given query coordinates X, it directly predicts $\hat{\mathbf{I}}^{HR-View}$ from $\hat{\mathbf{I}}^{LR-ERP}$ without the need to produce an HR ERP image.

$\mathbf{X}_{view}$ from the latent space of input $\hat{\mathbf{I}}^{LR-ERP}$ instead of using traditional interpolation methods. As depicted in Fig. 3, our VR module consists of an encoder $\mathcal{E}$, a local texture estimator $h_\psi = \{h_a, h_f, h_p\}$, and an MLP decoder $\mathcal{D}$. $h_\psi$ is a learnable dominant-frequency estimator, which is capable of characterizing image textures in 2D Fourier space [23]. Here, $h_a$ is an amplitude estimator ($\mathbb{R}^C \mapsto \mathbb{R}^{256}$), $h_f$ is a frequency estimator ($\mathbb{R}^C \mapsto \mathbb{R}^{2\times128}$), and $h_p$ is a phase estimator ($\mathbb{R}^{10} \mapsto \mathbb{R}^{128}$). Concretely, for a query point $\mathbf{x} \in \mathbf{X}_{view}$, the estimating function $h_\psi$ is defined as:

$$h_\psi(\mathbf{z}_j, \delta_\mathbf{x}, \mathbf{s}(\mathbf{x})) = h_a(\mathbf{z}_j) \otimes \begin{bmatrix} \cos\{\pi(< h_f(\mathbf{z}_j), \delta_\mathbf{x} > +h_p(\mathbf{s}(\mathbf{x})))\} \\ \sin\{\pi(< h_f(\mathbf{z}_j), \delta_\mathbf{x} > +h_p(\mathbf{s}(\mathbf{x})))\} \end{bmatrix},$$
(1)

where $\mathbf{z} = \mathcal{E}(\hat{\mathbf{I}}^{LR-ERP})$. Denote $j$ as a pixel index of the $\hat{\mathbf{I}}^{LR-ERP}$, $\mathbf{x}_j$ and $\mathbf{z}_j$ are the corresponding coordinates and latent variable of $\hat{\mathbf{I}}^{LR-ERP}$, respectively. $\delta_\mathbf{x}$ is a local grid calculated by $\delta_\mathbf{x} = \mathbf{x} - \mathbf{x}_j$. $<\cdot, \cdot>$ is an inner product, and $\otimes$ denotes element-wise multiplication. To be noted, $\mathbf{s}(\mathbf{x})$ is the spherical pixel shape representation of the query coordinate $\mathbf{x}$, which is important for providing spatial-varying priors [22] for our VR module and will be elaborated on in Sec. 3.4. Finally, our VR module predicts the RGB values of a

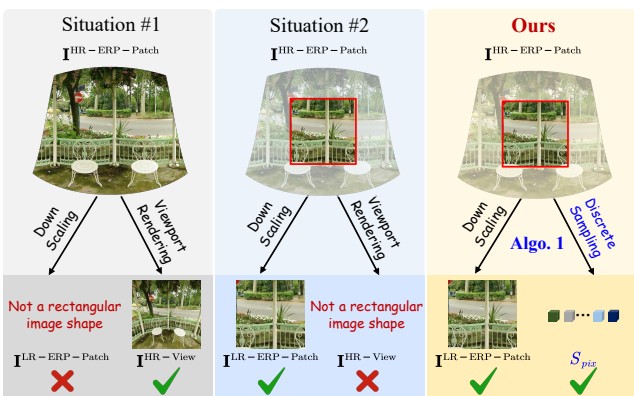

Figure 4: Training ResVR end-to-end faces challenges due to the mismatch in shapes between the ERP image patch ($\mathbf{I}^{HR-ERP-Patch}$) and the viewport ($\mathbf{I}^{HR-View}$). In Situation #1, although we obtain $\mathbf{I}^{HR-View}$ with a rectangular image shape, its corresponding $\mathbf{I}^{LR-ERP-Patch}$ does not have a rectangular image shape, preventing its use in supervising the downscaling process. Situation #2 experiences the opposite issue. Both two situations are impractical for training. In contrast, our method utilizes a novel discrete pixel sampling strategy (Algo. 1) to make end-to-end training feasible.

coordinate $\mathbf{y} = f(\mathbf{x})$ on the viewport as:

$$\hat{\mathbf{I}}^{HR-View}[\mathbf{y}] = \mathcal{F}_B(\hat{\mathbf{I}}^{LR-ERP}) + \sum_{j\in\mathcal{J}} \omega_j \mathcal{D}(h_\psi(\mathbf{z}_j, \delta_\mathbf{x}, \mathbf{s}(\mathbf{x}))), \quad (2)$$

where $\mathcal{F}_B$ is a bilinear interpolation operator to stabilize the convergence of network training and aid the VR module in learning high-frequency details. $\mathcal{J}$ is a neighborhood set of $\mathbf{x}$, defined as $\mathcal{J} = \{j|j = \mathbf{x} + [\frac{ms}{W}, \frac{ns}{H}], m, n \in \{-1, 1\}\}$, while $\omega_j$ is a local ensemble coefficient. More details of our VR module are provided in the supplementary material. Different from existing ODI rescaling methods which first produce $\hat{\mathbf{I}}^{HR-ERP}$ and then obtain HR viewports through traditional interpolation methods, our ResVR directly renders the final HR viewports from $\hat{\mathbf{I}}^{LR-ERP}$ through our VR module.

## 3.3 Discrete Pixel Sampling Strategy

**Motivation.** In image rescaling and SR tasks, the ground truth HR images $\mathbf{I}^{HR-ERP}$ are typically cropped into patches $\mathbf{I}^{HR-ERP-Patch}$ to reduce GPU memory overhead while increasing the diversity of training data. The goals of our ResVR's end-to-end training process include: (1) optimizing the downscaling process ($\mathbf{I}^{HR-ERP-Patch} \mapsto \hat{\mathbf{I}}^{LR-ERP-Patch}$), and (2) optimizing the rendering process ($\hat{\mathbf{I}}^{LR-ERP-Patch} \mapsto \hat{\mathbf{I}}^{HR-View}$). In this framework, the former process requires the corresponding $\mathbf{I}^{LR-ERP-Patch}$ as supervision, while the latter process needs the corresponding $\mathbf{I}^{HR-View}$ as supervision. However, due to the natural geometric properties of ODIs' different projection types, there exists a challenge of shape mismatch between a viewport and its corresponding ERP area. As depicted in Situations #1 and #2 in Fig. 4, one can not obtain $\mathbf{I}^{LR-ERP-Patch}$ and $\mathbf{I}^{HR-View}$ that are both with rectangular image shape at the same time. As a result, conventional image-level loss can not be directly used for end-to-end training of ResVR. To address this, we innovatively propose a discrete pixel sampling strategy (DPS), as shown in "Ours" of Fig. 4.

**Algorithm 1** Discrete pixel sampling strategy

$\mathbf{I}^{\text{HR-ERP}}$, $s$: HR ERP image and the downscaling factor
$(a, b)$, $p$: left top coordinate and the size of cropped patch
$(\theta_c, \phi_c)$, $(F_h, F_v)$: view direction and FoVs of the viewport
$(h_v, w_v)$, $\mathbf{Y}_{view}$: shape and the coordinate space of the viewport

> **function** DisSamp($a, b, p, \theta_c, \phi_c, F_h, F_v, h_v, w_v, \mathbf{Y}_{view}, \mathbf{I}^{\text{HR-ERP}}$)
>   $\mathbf{I}^{\text{HR-ERP-Patch}} \leftarrow$ CropPatch($\mathbf{I}^{\text{HR-ERP}}, a, b, p$)
>   $f \leftarrow$ GetTransform($\theta_c, \phi_c, F_h, F_v, h_v, w_v$)
>   $\mathbf{X}_{view} \leftarrow f^{-1}(\mathbf{Y}_{view})$     ▷ Inverse mapping from viewport to ERP
>   $\mathbf{X}'_{view} \leftarrow$ FilterWithBounds($\mathbf{X}_{view}, a, b, p$)     ▷ Eq. (3)
>   **if** $|\mathbf{X}'_{view}| > N$ **then**
>     $\mathbf{X}'_{view} \leftarrow$ RandomSample($\mathbf{X}'_{view}, N$)
>   **end if**
>   $\mathbf{X}_{samp} \leftarrow$ CoordSpaceTransform($\mathbf{X}'_{view}, a, b, p$)     ▷ Eq. (4)
>   $S_{pix} \leftarrow$ BicubicSample($\mathbf{I}^{\text{HR-ERP-Patch}}, \mathbf{X}_{samp}$)
>   $\mathbf{I}^{\text{LR-ERP-Patch}} \leftarrow$ BicubicDownscale($\mathbf{I}^{\text{HR-ERP-Patch}}, s$)
>   **return** $\mathbf{X}_{samp}, S_{pix}, \mathbf{I}^{\text{LR-ERP-Patch}}$
> **end function**

**Method.** Denoting $p$ as both the height and width of training patches, our core idea is to keep the LR ERP patch with a rectangular image shape ($\mathbf{I}^{\text{LR-ERP-Patch}} \in \mathbb{R}^{3 \times \frac{p}{s} \times \frac{p}{s}}$), while discretely sampling $N$ pixels $S_{pix} \in \mathbb{R}^{3 \times N}$ with their coordinates $\mathbf{X}_{samp} \in \mathbb{R}^{2 \times N}$ in the irregular area. Thanks to the continuous representation ability of implicit neural representation methods [11], our VR module is able to predict corresponding pixel values $\hat{S}_{pix}$ given coordinates $\mathbf{X}_{samp}$. Hence, we use the sampled $S_{pix}$ as supervision for the predicted $\hat{S}_{pix}$. An overview of our strategy is presented in Algo. 1.

Specifically, given the left top coordinate $(a, b)$ of the training patch, the view direction $(\theta_c, \phi_c)$, FoVs ($F_h$ and $F_v$), shapes ($h_v$, $w_v$), and query coordinates ($\mathbf{Y}_{view}$) of the desired viewport, we first determine the coordinate mapping $f$. Then the corresponding coordinates $\mathbf{X}_{view}$ in the ERP space can be obtained through inverse coordinate mapping as $\mathbf{X}_{view} = f^{-1}(\mathbf{Y}_{view})$. By setting appropriate view direction and FoVs, it is possible to ensure that some elements in $\mathbf{X}_{view}$ lie in the area of the cropped ERP patch. Recall that $p$ is the size of the cropped training patch, $\mathbf{X}_{view}$ is then filtered to preserve the subset overlapped with the patch as follows:

$$\mathbf{X}'_{view} = \{\mathbf{x} \in \mathbf{X}_{view} | a \le \mathbf{x}_1 < (a + p), b \le \mathbf{x}_2 < (b + p)\}, \quad (3)$$

where $\mathbf{x}_1$ and $\mathbf{x}_2$ represent the coordinates of a point $\mathbf{x}$ in the ERP space. This filter operation ensures that every point in $\mathbf{X}'_{view}$ can find its corresponding latent variable extracted from $\hat{\mathbf{I}}^{\text{LR-ERP-Patch}}$. To balance the number of pixels sampled from different FoVs and view directions, if the filtered $\mathbf{X}'_{view}$ contains more than $N$ pixel coordinates, only $N$ elements will be randomly retained. Finally, to uniform the coordinate correspondence between LR and HR patches, $\mathbf{X}'_{view} \subseteq [0, H] \times [0, W]$ is converted from the whole HR ERP coordinate space to the patch coordinate space, and normalized to $\mathbf{X}_{samp} \subseteq [-1, 1) \times [-1, 1)$, as:

$$\mathbf{X}_{samp} = \{T(\mathbf{x}) | \mathbf{x} \in \mathbf{X}'_{view}\},$$
$$\text{where } T(\mathbf{x}) = 2\left(((\mathbf{x}_1, \mathbf{x}_2) - (a, b))/p\right) - 1. \quad (4)$$

As a result, $S_{pix}$ are sampled through bicubic interpolation as $S_{pix} = \mathcal{F}_{bicubic}(\mathbf{I}^{\text{HR-ERP-Patch}}, \mathbf{X}_{samp})$. By adopting Algo. 1, $\mathbf{I}^{\text{LR-ERP-Patch}} \in$

$\mathbb{R}^{3 \times \frac{p}{s} \times \frac{p}{s}}$ can be used as supervision of the predicted $\hat{\mathbf{I}}^{\text{LR-ERP-Patch}}$, and $S_{pix} \in \mathbb{R}^{3 \times N}$ can be used as supervision of the predicted $\hat{S}_{pix}$.

## 3.4 Spherical Pixel Shape Representation

**Motivation.** Pixel shape representations described by grid orientation and curvature provide informative geometric spatial-varying priors for image warping tasks [22]. Existing shape representation methods are designed for 2D images. However, different from 2D images, ERP is an unfolding of the spherical surface along meridians, which leads to the fact that the adjacent pixels on the sphere can be far apart in the ERP image. As a result, the previous 2D shape representation methods fall in the high-latitude/-longitude areas due to the nature properties of (1) latitude-related distortion and (2) wraparound consistency of ERP images (Fig. 7 in Sec. 4.3). Therefore, we develop a spherical pixel shape representation (SSR) technique to solve this challenge. As Fig. 5 shows, SSR leverages the information of transformed coordinates on the sphere as a more effective shape representation to guide the viewport rendering process.

**Method.** In image warping tasks, the first-order partial derivatives (i.e. Jacobian matrix) and the second-order partial derivatives (i.e. Hessian matrix), describe the orientation and curvature of pixels resulting from the transformation, respectively [22]. These two matrices provide informative geometric spatial-varying priors to guide the process of viewport rendering. Inspired by this, for a point $\mathbf{x}$ on the original ERP plane, we represent its pixel shape $\mathbf{s}(\mathbf{x})$ with the gradient of the inverse coordinate transformation $f^{-1}$. Specifically, we start with the point $\mathbf{y} = f(\mathbf{x})$ on the viewport plane, and the corresponding Jacobian matrix $\widetilde{\mathbf{J}}_{f^{-1}}(\mathbf{y})$ and Hessian matrix $\widetilde{\mathbf{H}}_{f^{-1}}(\mathbf{y})$ are analytically computed as:

$$\widetilde{\mathbf{J}}_{f^{-1}}(\mathbf{y}) = \begin{bmatrix} \dfrac{\partial \mathbf{x}_1}{\partial u} & \dfrac{\partial \mathbf{x}_1}{\partial v} \\ \dfrac{\partial \mathbf{x}_2}{\partial u} & \dfrac{\partial \mathbf{x}_2}{\partial v} \end{bmatrix}, \quad \widetilde{\mathbf{H}}_{f^{-1}}(\mathbf{y}) = \begin{bmatrix} \dfrac{\partial^2 \mathbf{x}_1}{\partial^2 u} & \dfrac{\partial^2 \mathbf{x}_1}{\partial u \partial v} \\ \dfrac{\partial^2 \mathbf{x}_2}{\partial v \partial u} & \dfrac{\partial^2 \mathbf{x}_2}{\partial^2 v} \end{bmatrix}, \quad (5)$$

where $\mathbf{x} = f^{-1}(\mathbf{y}) = (\mathbf{x}_1, \mathbf{x}_2)$ represents the coordinates in the original ERP plane. We propose a spherical pixel shape representation (SSR) technique to numerically estimate $\widetilde{\mathbf{J}}_{f^{-1}}(\mathbf{y})$ and $\widetilde{\mathbf{H}}_{f^{-1}}(\mathbf{y})$ based on spherical differentiation. Specifically, as depicted in Fig. 5, the inverse coordinate transformation $f^{-1}$ is firstly applied to $\mathbf{y}$ and its eight nearest points ($\mathbf{y} + [\frac{m}{w_v}, \frac{n}{h_v}]$ with $m, n \in \{-1, 0, 1\}$) to get $\mathbf{x}$ and its neighborhood on the ERP plane. Different from existing 2D shape representation methods [22] which estimate the shape of $\mathbf{x}$ directly on the ERP image plane, we further transform $\mathbf{x}$ and its neighborhood to the sphere denoted by the set $\{\mathbf{p}_i | \mathbf{p}_i = (\theta_i, \phi_i), i = 1, 2, \dots, 9\}$, and innovatively calculate numerical derivatives to estimate $\widetilde{\mathbf{J}}_{f^{-1}}(\mathbf{y})$ and $\widetilde{\mathbf{H}}_{f^{-1}}(\mathbf{y})$ by the proposed spherical central difference method, overriding Eq. (5) as:

$$\widetilde{\mathbf{J}}_{f^{-1}}(\mathbf{y}) \approx \begin{bmatrix} D(\mathbf{p}_6, \mathbf{p}_4) \\ D(\mathbf{p}_2, \mathbf{p}_8) \end{bmatrix},$$
$$\widetilde{\mathbf{H}}_{f^{-1}}(\mathbf{y}) \approx \begin{bmatrix} D(\mathbf{p}_6, \mathbf{p}_5) + D(\mathbf{p}_5, \mathbf{p}_4) & D(\mathbf{p}_3, \mathbf{p}_1) + D(\mathbf{p}_9, \mathbf{p}_7) \\ D(\mathbf{p}_3, \mathbf{p}_1) + D(\mathbf{p}_9, \mathbf{p}_7) & D(\mathbf{p}_2, \mathbf{p}_5) + D(\mathbf{p}_5, \mathbf{p}_8) \end{bmatrix}, \quad (6)$$

**Figure 5: Illustration of our proposed spherical pixel shape representation (SSR) technique. We illustrate using a point y on the viewport. The inverse mapping is firstly applied for y and its eight nearest neighbors to get x and its neighbors on ERP. Then these points are transformed into sphere coordinates $\{p_1, p_2, \cdots, p_9\}$, which are used for calculating numerical derivatives to estimate the pixel shape representation s(x), according to proposed spherical central difference method in Eqs. (6) and (7).**

where $D(\mathbf{p}_i, \mathbf{p}_j)$ calculates the distance between $\mathbf{p}_i$ and $\mathbf{p}_j$ in the spherical coordinate system as:

$$D(\mathbf{p}_i, \mathbf{p}_j) = \begin{pmatrix} \min(|\phi_j - \phi_i|, 2\pi - |\phi_j - \phi_i|) \\ \min(|\theta_j - \theta_i|, 2\pi - |\theta_j - \theta_i|) \cdot \cos\left((\theta_i + \theta_j)/2\right) \end{pmatrix}^\top, \quad (7)$$

where $|\cdot|$ represents the absolute value operation. The $\min(\cdot)$ operator ensures the computation of minor arc distances, and the average latitude is utilized to adjust the longitudinal distance to account for the convergence of meridians. We follow [22] to use six elements in $\widetilde{\mathbf{H}}_{f^{-1}}(\mathbf{y})$, and finally $s(\mathbf{x}) \in \mathbb{R}^{10}$ is obtained by concatenating and flattening $\widetilde{\mathbf{J}}_{f^{-1}}(\mathbf{y})$ and $\widetilde{\mathbf{H}}_{f^{-1}}(\mathbf{y})$. As a result, $s(\mathbf{x})$ serves as an auxiliary input, together with the LR latent representation $\mathbf{z} = \mathcal{E}(\hat{\mathbf{I}}^{\text{LR-ERP}})$ to predict final viewports through the VR module.

## 3.5 Training Objectives

Thanks to the discrete pixel sampling strategy (Sec. 3.3) and the fully differentiable pipeline (Fig. 2), the processes of downscaling, compression, and viewport rendering can be jointly optimized end-to-end, which aligns the learning of network parameters with our goals of obtaining high-quality viewport while reducing the transmission overhead. The total loss is a weighted sum of a pixel-level reconstruction loss, an LR guidance loss, and a bitrate loss as:

$$\mathcal{L} = \mathcal{L}_{\text{pix}} + \lambda_1 \mathcal{L}_{\text{guide}} + \lambda_2 \mathcal{L}_{\text{bpp}}, \quad (8)$$

where $\lambda_1$ and $\lambda_2$ are two trade-off parameters.

**Pixel-level reconstruction and guidance loss.** In the training phase, we employ Algo. 1 to get $S_{pix}$ as the ground truth and use the VR module to predict the corresponding values $\hat{S}_{pix}$ given $X_{samp}$.

$$\mathcal{L}_{\text{pix}} = \frac{||\hat{S}_{pix} - S_{pix}||_1}{N}, \quad (9)$$

where $N$ is the number of sampled pixels. Additionally, following [35, 46, 47], an $L_2$ guidance loss on the LR ERP patch is defined as:

$$\mathcal{L}_{guide} = \frac{||\hat{\mathbf{I}}^{\text{LR-ERP-Patch}} - \mathbf{I}^{\text{LR-ERP-Patch}}||_2^2}{(p/s) \times (p/s)}. \quad (10)$$

**Bitrate loss.** To optimize the size of the transmitted JPEG image, we firstly follow [3] to estimate the rate $R$ of the quantized coefficients $\widetilde{C}$ with differentiable fully factorized entropy models as: $R = \mathbb{E}_{x \sim p_x}[-\log p_L(\widetilde{C}_y) - \log p_C(\widetilde{C}_{Cb}) - \log p_C(\widetilde{C}_{Cr})]$, where $p_L$ and $p_C$ are two fully factorized entropy models for luma and chroma

coefficient maps, respectively. Then the bpp of the transmitted LR ERP JPEG image is calculated as:

$$\mathcal{L}_{bpp} = \frac{R}{H \times W}. \quad (11)$$

## 4 EXPERIMENTS

### 4.1 Experimental Setup

**Implementation details.** We follow the previous work [35] to put more computation into the downsampler to keep the VR module lightweight. The VR module is composed of a lightweight feature extractor and a tiny MLP. Please refer to our supplementary materials for more details of our network architecture.

**Training details.** The downscaling factor $s$ is fixed to 4, and patch size $p$ of random cropped $\mathbf{I}^{\text{HR-ERP-Patch}}$ is set to 256. To ensure the overlapping of $\mathbf{I}^{\text{HR-ERP-Patch}}$ and discretely sampled points in Algo. 1, we calculate the viewport center $(\theta_c, \phi_c)$ according to the center of cropped patch. To enable ResVR to handle different resolutions and FoVs, we randomly sample FoVs and resolutions during training. Specifically, the FoVs $(F_h, F_v)$ are randomly sampled from $\{80°, 90°, 100°, 110°, 120°\}$ and the resolutions of viewport $(h_v, w_v)$ are randomly sampled from $\{512, 576, 640, 768, 832, 960, 1024\}$. The number of sampled pixels $N$ is set to 25600, and the batch size is set to 16. All experiments are conducted on one V100 GPU. The network is trained for $5 \times 10^5$ iterations with learning rate $2 \times 10^{-4}$. $\lambda_1$ and $\lambda_2$ are set to 0.6 and 0.01 for all experiments, respectively.

**Datasets and evaluation metrics.** ODI-SR dataset [14] and SUN360 Panorama dataset [45] are used in our experiment. We follow the data split setting in [14] and train on the ODI-SR training set. For evaluation of transmission efficiency, we follow [35] to use the real file size of JPEG for evaluating the bitrate: bpp = $\mathbb{E}_{x \sim p_x}[\text{filesize}/(H \times W)]$. For evaluation of the quality of rendered viewports, we choose ten different view directions, get the ground truth image using bicubic interpolation, and calculate the average PSNR, SSIM, and LPIPS [59]. More details are provided in the supplementary materials. For those competing methods that need to explicitly produce HR ERP images to render final viewports, we evaluate the WS-PSNR [40] of their predicted $\hat{\mathbf{I}}^{\text{HR-ERP}}$.

### 4.2 Comparison with State-of-the-Art Methods

We compare three categories of methods: (1) a baseline method which downscales with Bicubic interpolation, compresses with standard JPEG codec, and renders viewports with Bicubic interpolation;

**Table 1: Quantitative evaluation of rendered viewports with $(F_h, F_v) = (120°, 90°)$ and $(w_v, h_v) = (2048, 1536)$. We keep bpp around 0.3 on different datasets. The WS-PSNR is evaluated for methods that need to explicitly get HR ERP to render the viewport. Focusing solely on the quality of ERP images results in the sub-optimal visual experience of final viewports. Throughout this paper, the best and second-best results of each test setting are highlighted in bold red and underlined blue, respectively.**

| Method | ODI-SR [14] | | | | | SUN 360 [45] | | | | |
|---|---|---|---|---|---|---|---|---|---|---|
| Down & Compression & Up & Render | bpp↓ | WS-PSNR | PSNR↑ | SSIM↑ | LPIPS↓ | bpp↓ | WS-PSNR | PSNR↑ | SSIM↑ | LPIPS↓ |
| Bicubic & JPEG & Bicubic | 0.29 | N/A | 27.98 | 0.7897 | 0.4815 | 0.28 | N/A | 28.06 | 0.8077 | 0.4642 |
| Bicubic & JPEG & OSRT [58] & Bicubic | 0.29 | 25.73 | 28.42 | 0.7975 | 0.4395 | 0.28 | 26.09 | 28.84 | 0.8202 | 0.5510 |
| HyperThumbnail [35] & Nearest | 0.30 | 27.84 | 30.13 | 0.8314 | 0.4197 | 0.29 | 28.97 | 31.16 | 0.8601 | 0.3768 |
| HyperThumbnail [35] & Bilinear | 0.30 | 27.84 | 30.82 | 0.8475 | 0.3397 | 0.29 | 28.97 | 32.20 | 0.8775 | 0.2792 |
| HyperThumbnail [35] & Bicubic | 0.30 | 27.84 | _31.00_ | _0.8515_ | _0.3222_ | 0.29 | 28.97 | _32.54_ | _0.8822_ | _0.2610_ |
| **ResVR (Ours)** | 0.30 | N/A | **31.39** | **0.8568** | **0.3026** | 0.29 | N/A | **32.95** | **0.8862** | **0.2462** |

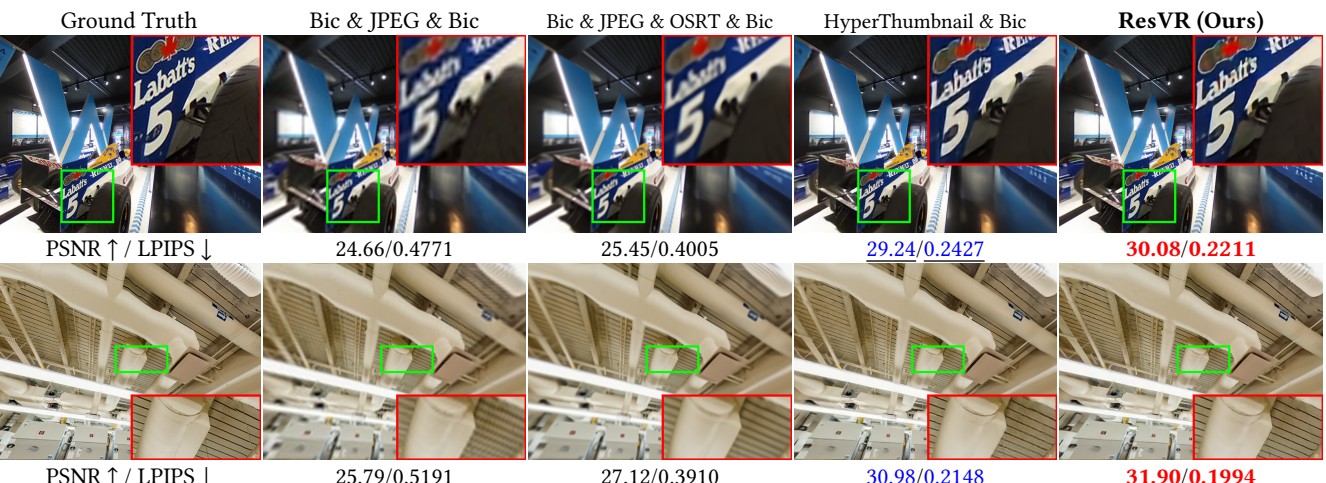

| Ground Truth | Bic & JPEG & Bic | Bic & JPEG & OSRT & Bic | HyperThumbnail & Bic | **ResVR (Ours)** |
|---|---|---|---|---|
| PSNR ↑ / LPIPS ↓ | 24.66/0.4771 | 25.45/0.4005 | _29.24_/_0.2427_ | **30.08**/**0.2211** |
| PSNR ↑ / LPIPS ↓ | 25.79/0.5191 | 27.12/0.3910 | _30.98_/_0.2148_ | **31.90**/**0.1994** |

**Figure 6: Comparisons of two rendered viewports from ODI-SR [14] ("img_005", top) and SUN 360 [45] ("img_046", bottom), with $(\theta = 90°, \phi = 0°)$ and $(\theta = 180°, \phi = 45°)$, respectively. The viewports are with FoVs $(F_h, F_v) = (120°, 90°)$ and resolutions $(w_v, h_v) = (2048, 1536)$. "Bic" stands for Bicubic interpolation. Please zoom in for more details.**

(2) SR pipeline which downscales with Bicubic, compresses with standard JPEG codec, upscales with state-of-the-art methods [58], and uses bicubic interpolation for viewport rendering; (3) Rescaling pipeline which firstly uses state-of-the-art asymmetric rescaling methods [35] and renders viewports with Bicubic interpolation. For a fair comparison, we retrain SR methods [58] and rescaling methods [35] with our training sets and constrain their bpp around 0.3 by adjusting the quality factor of JPEG compression in all baselines.

Tab. 1 presents the quantitative comparisons of the quality of rendered viewports with $(F_h, F_v) = (120°, 90°)$ and $(w_v, h_v) = (2048, 1536)$ among different methods. Taking advantage of directly optimizing the final viewport through end-to-end training, ResVR outperforms previous methods by reconstruction accuracy (about 0.4dB gain on PSNR) and with better realness (about 0.02 gain on LPIPS). Notably, even though existing rescaling methods can achieve PSNR of about 28dB, they only obtain sub-optimal results due to the lack of awareness and optimization of the viewport rendering process, which demonstrates the fact that focusing solely on the quality of ERP images results in sub-optimal visual experience of viewports. ResVR directly optimizes the rendered viewports without the need to predict HR ERP and achieves SOTA performance. Fig. 6 further provides the visual comparison of rendered viewports.

It can be seen that our ResVR reconstructs enhanced details with fewer artifacts (e.g. the text "labatt's" in the top image and the lines in the bottom image) compared to other methods. Additionally, ResVR shows better wraparound continuity at the location of the seams (e.g., pipe in the zoomed-in area) in the bottom case. We attribute this to the awareness of arbitrary view directions during training and our MLP's continuous representation of images.

### 4.3 Ablation Study

In this section, we analyze the effect of the proposed discrete pixel sampling strategy and spherical pixel shape representation. We conduct experiments on three ResVR variants: (1) Case #1: We train the rescaling process and VR module separately with 2D shape representation; (2) Case #2: We train the whole pipeline in an end-to-end manner by employing discrete pixel sampling strategy with 2D shape representation; (3) The complete ResVR with our sphere pixel shape representation. Quantitative results are shown in Tab. 2.

**Effect of discrete pixel sampling strategy.** Comparison #1 vs. #2 exhibits the effectiveness of the proposed discrete pixel sampling strategy which enables end-to-end training of the whole pipeline, thus providing substantial PSNR improvement of 4.12-4.44dB. To

**Table 2: Ablation study on proposed discrete pixel sampling strategy (DPS) and spherical pixel shape representation (SSR) with setting $(F_h, F_v) = (90°, 90°)$ and $(w_v, h_v) = (1024, 1024)$.**

| Case | Test Set | DPS | SSR | PSNR↑ | SSIM↑ | LPIPS↓ |
|------|----------|-----|-----|-------|-------|--------|
| #1 | ODISR | ✘ | ✘ | 27.35 | 0.8296 | 0.3321 |
| #2 | | ✔ | ✘ | 31.47 | 0.8560 | 0.2820 |
| **ResVR** | | ✔ | ✔ | **31.87** | **0.8606** | **0.2696** |
| #1 | SUN360 | ✘ | ✘ | 28.09 | 0.8503 | 0.3040 |
| #2 | | ✔ | ✘ | 32.53 | 0.8742 | 0.2443 |
| **ResVR** | | ✔ | ✔ | **33.08** | **0.8793** | **0.2324** |

| Ground Truth | Case #2 (w/o SSR) | ResVR (Ours) |
|---|---|---|
| $\theta = 0°, \phi = 0°$ | 38.70/0.1892 | 38.97/0.1865 |
| $\theta = 0°, \phi = -90°$ | 34.48/0.3103 | 39.19/0.2554 |
| $\theta = 180°, \phi = 0°$ | 36.89/0.2188 | 37.79/0.2037 |

**Figure 7: Visual comparison of two variants (Case #2 and Ours). 2D shape representation falls in high-latitude and high-longitude areas (highlighted by red arrow). In contrast, our spherical representation ensures stable rendering results in various directions with better PSNR (dB)↑ and LPIPS ↓.**

further analyze the sampled coordinates and pixels, we visualize the sampled areas and pixels in the supplementary materials.

**Effect of spherical pixel shape representation.** Comparison #2 vs. ResVR reveals that the spherical shape representation improves PSNR by 0.40-0.55dB. We observe that conventional 2D shape representation methods [22] fall in high-longitude and high-latitude areas, as depicted in Fig. 7. We attribute this to the following reasons: adjacent pixels on the viewport should be very close to each other on the sphere. However, (1) for high-latitude areas, due to the natural distortion of ERP, the distance between corresponding points is very close to the original sphere but is very far on the ERP image. (2) For high-longitude areas, due to the wraparound consistency of the left and right ends of ERP, the corresponding

**Table 3: Quantitative evaluation on viewports of different resolutions with $(F_h, F_v) = (120°, 120°)$. ResVR outperforms existing methods under various resolutions. Bic: Bicubic interpolation, Res: Resolution, HT: HyperThumbnail [35].**

| Method | Res | ODI-SR [14] | | | SUN 360 [45] | | |
|--------|-----|-------|-------|--------|-------|-------|--------|
| | | PSNR↑ | SSIM↑ | LPIPS↓ | PSNR↑ | SSIM↑ | LPIPS↓ |
| HT [35] & Bic | $512^2$ | 30.89 | 0.8704 | 0.1976 | 32.08 | 0.8880 | 0.1630 |
| **ResVR (Ours)** | | **31.03** | **0.8754** | **0.1888** | **32.16** | **0.8914** | **0.1527** |
| HT [35] & Bic | $1024^2$ | 30.92 | 0.8498 | 0.2802 | 32.09 | 0.8718 | 0.2343 |
| **ResVR (Ours)** | | **31.25** | **0.8579** | **0.2632** | **32.39** | **0.8782** | **0.2176** |
| HT [35] & Bic | $2048^2$ | 30.93 | 0.8541 | 0.3230 | 32.08 | 0.8769 | 0.2713 |
| **ResVR (Ours)** | | **31.32** | **0.8596** | **0.3046** | **32.48** | **0.8812** | **0.2567** |

points are originally adjacent pixels on the sphere but are at both ends of the ERP image. The traditional shape representation method based on 2D ERP ignores the spherical characteristics, so in the above situations, it provides wrong pixel shape priors for the VR module, thus leading to abnormal visual results. The proposed SSR is performed on the native spherical surface, thus ensuring stable and high-quality rendering quality in various view directions.

## 4.4 Analysis

**LR JPEG image.** The file size and visual quality of LR JPEG images are also important for efficient transmission and user preview. ResVR effectively balances transmission efficiency with the final viewer experience through our end-to-end training. Visualizations of LR images are provided in the supplementary materials.

**Arbitrary-resolution viewport rendering.** Due to the continuous representation ability [11, 22, 23] of the MLP in our VR module, ResVR is able to render viewports of different resolutions using only one set of parameters. Tab. 3 compares ResVR with SOTA methods on different resolution settings. It can be seen that ResVR achieves better performance at different resolutions.

## 5 CONCLUSION

The rise of virtual reality and augmented reality applications has popularized ODI rescaling to shrink the file size of images while maintaining their quality. However, existing methods focus on improving the ERP image quality, ignoring that HMDs use rendered viewports for display, not ERP images. This focus on ERP quality alone will lead to compromised user experiences. To address this, in this paper, we propose ResVR, a novel framework for joint rescaling and viewport rendering of ODIs. In ResVR, we develop a discrete pixel sampling strategy to tackle the irregular correspondence between the ERP area and the viewport, enabling end-to-ending training of the whole pipeline. A spherical pixel shape representation technique is introduced to serve as an effective guidance for the rendering process, further enhancing the visual quality of viewports. Experiments demonstrate that our ResVR achieves state-of-the-art performance across different settings of FoVs, view directions, and resolutions while keeping a low transmission overhead.

**Limitations and future work.** Existing ODI datasets are of relatively low resolution and with compression artifacts, which limits the further performance improvements of ResVR. Besides, although ResVR provides a promising pipeline for comprehensive ODI processing and achieves SOTA performance, it is crucial to extend ResVR to 360° video tasks. We leave this for future work.

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
