# OpenReview forum: "ResVR: Joint Rescaling and Viewport Rendering of Omnidirectional Images"
_acmmm.org/ACMMM/2024/Conference — MM2024 Oral_

### Official Review · Reviewer_kv6y · 2024-05-17

**Rating:** 4
**Confidence:** 3

**Summary:**

The paper addresses the challenges related to bandwidth requirement for an omnidirectional contents (ODI images). Since many applications downscale the images during transmission to save bandwidth and then upscale it at user's end, the quality of content gets degraded.
The proposed paper presents a method, which instead of upscaling the whole ODI, only focuses on the viewport rendering part, means the planar patch of image which is currently being viewed by the user. The pipeline works by first down sampling and compression of the ODI image (in ERP format). This image is then decompressed, and then up sampled for the target viewport by using a learning based up sampling module.
In the comparison the method achieves significant performance in terms of visual quality when compared to the other mentioned methods.

**Strengths:**

**1**. The rescaling and viewport rendering in single pipeline ensures optimization is directly aligned with the final visual output on head-mounted displays (HMDs).

**2**. The pixel sampling addresses the irregular correspondence issues inherent in spherical images, enabling more accurate and high-quality reconstructions.

**3**. Spherical pixel shape representation technique gives better geometric orientation and curvature information, particularly improving visual quality in regions with high latitude and longitude variations, which are typically problematic in conventional methods

**4**. The proposed method achieves state-of-the-art performance in multiple viewport rendering tasks. This validation across different fields of view, resolutions, and view directions highlights the robustness and versatility of the proposed framework.

**Limitations:**

**1.** The paper focuses on the visual quality of the rendered FoV, however given the final application viewing a 360 image, where the user might move the head randomly at different latitudes and longitudes. The runtime of model becomes significant. The inference time should have been mentioned in the table 1,2 and 3.

**2.** The paper does not mention about the scalability of the ResVR framework in real-time applications. Given the high computational demands, there may be issues related to latency and performance, especially under varying network conditions and hardware capabilities that are typical in practical VR scenarios.

**3.** For comparison the paper uses **[58]Yu, F., Wang, X., Cao, M., Li, G., Shan, Y. and Dong, C., 2023. Osrt: Omnidirectional image super-resolution with distortion-aware transformer. In Proceedings of the IEEE/CVF Conference on Computer Vision and Pattern Recognition (pp. 13283-13292)**. However considering the planar nature of final viewport image, addition of a planar image SR method will have added more weights to the results **Liang, J., Cao, J., Sun, G., Zhang, K., Van Gool, L. and Timofte, R., 2021. Swinir: Image restoration using swin transformer. In Proceedings of the IEEE/CVF international conference on computer vision (pp. 1833-1844).**



**Minor Notes:**
In section 4.1 (experimental setup), the line 676, it mentions about the number of iterations the model was trained for. Since these iterations will change according to data points and batch size. It will be better to mention about number of epochs (even though authors have mentioned about these 2 values)

**Suitability:**

3

---

### Official Review · Reviewer_JV9Q · 2024-05-21

**Rating:** 5
**Confidence:** 3

**Summary:**

This paper proposes a rescaling with viewport rendering SR model for omnidirectional images (ODI). These rescaling models can significantly reduce the burden of processing by compressing low-resolution images with a conventional JPEG codec [1]. Overall, this paper clearly describes the proposed method and the performance improvement is also substantial. In addition, the comparison of experimental results is solid, and additional visualization materials also explain the proposed techniques well.

**Strengths:**

In this paper, the authors achieved better performance for ODI SR task through 1) a discrete pixel sampling strategy (DPS) and 2) representation considering spherical coordinates.

A coordinate-grid based SR method, LTE is well-known but this paper effectively introduced it to the ERP rescaling task.

The DPS considering geometric properties demonstrates the potential to be widely applied across the ERP image task. Specifically, DPS was designed for effective learning within ERP datasets, taking viewport rendering into account. This method has shown significant performance improvements in experimental results. It seems valuable to introduce this model to the low-level vision task for ODI. The paper also demonstrates that SSR can effectively remove the artifacts from ERP images.

**Limitations:**

This paper did not compare with SOTA ERP SR model, DINN360 [2]. It seems that it was impossible to reproduce.

It would be good to describe the overall runtime of the proposed model. The comparison model, HyperThumbnail [1], explicitly states the runtime benefits through the JPEG codec. I think this paper is also expected to be similar.

[1] Qi, Chenyang, et al. "Real-time 6K Image Rescaling with Rate-distortion Optimization." Proceedings of the IEEE/CVF Conference on Computer Vision and Pattern Recognition. 2023.

[2] Guo, Yichen, et al. "Dinn360: Deformable invertible neural network for latitude-aware 360deg image rescaling." Proceedings of the IEEE/CVF Conference on Computer Vision and Pattern Recognition. 2023.

**Suitability:**

3

---

### Official Review · Reviewer_F9kL · 2024-05-24

**Rating:** 5
**Confidence:** 3

**Summary:**

This paper presents ResVR, a framework designed to optimize omnidirectional image processing for head-mounted displays by focusing on joint rescaling and viewport rendering. ResVR targets the actual viewports seen by users, improving both transmission efficiency and viewport image quality. It introduces a novel discrete pixel sampling strategy and a spherical pixel shape representation technique, enhancing the quality of rendered viewports.

**Strengths:**

The paper is technically sound, providing a clear description of the methods and algorithms used. The introduction of spherical pixel shapes derived from spherical differentiation to handle pixel distortions effectively is a significant technical contribution that improves the rendering quality of viewports;
Diagrams and comparisons are used effectively to illustrate the improvements ResVR brings and to explain the operational mechanisms of the framework;
The paper clearly outlines the implications of improved ODI processing for VR and AR applications.

**Limitations:**

Generally I think it's a very good paper.
Since the success of VR/AR applications heavily depends on user acceptance and satisfaction, the paper might not sufficiently explore the challenges of deploying the framework in actual virtual or augmented reality systems, such as integration issues with existing systems, user acceptance, and quality of experience in practical use.
What is the details in the downscaling stage? Maybe you did not consider the distortion in "polar" areas. Please check OSRT(https://arxiv.org/pdf/2302.03453) again for its Deep Feature Extraction which consider distortion in downsampling stage too besides only as a SR method in your paper. Thank you.

**Suitability:**

3

---

### Official Review · Reviewer_wHUF · 2024-05-27

**Rating:** 5
**Confidence:** 3

**Summary:**

In this paper, a learning-based framework for joint rescaling and viewport rendering of omnidirectional images is proposed. The paper assumes as a conventional pipeline that omnidirectional images are downscaled (LR version) and upscaled (HR version) in the ERP domain, followed by viewport extraction to render an HR view. The proposed approach instead relies on a learning-based module that extracts the viewport directly from the downscaled omnidirectional image (LR) to render an HR view. The main contributions of the paper include the discrete pixel sampling strategy and the spherical pixel shape representation technique.

**Strengths:**

+ well-written
+ solid contributions
+ technically sound
+ sufficient analysis

I find the paper quite good. The manuscript is well-written and well-structured, with clear motivations and contributions. Moreover, the proposed method seems technically sound, while the performance evaluation and ablation studies are adequate.

**Limitations:**

- missing complexity aspects

The main aspect I am missing is related to the complexity of the proposed approach during the inference stage. Specifically, there is no information on whether the proposed method can be used for real-time consumption of an omnidirectional image e.g., a user inspects the omnidirectional image through a head-mounted display and can choose the viewport in real time through head movements - is this feasible using the proposed method? What is the maximum/minimum update frame rate, assuming different viewports between successive frames? How does this compare to the conventional pipeline?

Cosmetic changes:
- "LR" is not defined as an abbreviation in the Abstract
- In section 4.2, it is written:
"(3) Rescaling pipeline which firstly uses state-of-the-art asymmetric rescaling methods [35] and renders viewports with Bicubic interpolation."
However, it seems that Nearest and Bilinear methods are additionally used in Table 1. Please clarify.
- It is not clear in Table 1 whether the average performance indexes are reported across all contents of a dataset, or something different. Moreover, it would be good to mention the number of contents/stimuli included in each selected dataset and potentially other characteristics.

**Suitability:**

3

---

### Meta-Review · Area_Chair_5coP · 2024-07-01

**Recommendation:** Accept (Oral)
**Confidence:** 4

**Metareview:**

The paper is clearly written.
It provides significant original contributions to the field, as appreciated by all four reviewers.
The less strong critical points / limitations have been addressed in the rebuttal, where possible in revision.

The positive and few critical points mentioned by the reviewers (summarized / partly cited from original reviews) are:

(wHUF)
-  well-written
-   solid contributions
-   technically sound
-   sufficient analysis

"...Moreover, the proposed method seems technically sound, while the performance evaluation and ablation studies are adequate."

(F9kL)
- technically sound,
- providing a clear description of the methods and algorithms used.
- spherical pixel shapes derived from spherical differentiation to handle pixel distortions effectively is a significant technical contribution that improves the rendering quality of viewports;
- diagrams and comparisons used effectively to illustrate improvements ResVR brings and to explain the operational mechanisms of the framework
- paper clearly outlines implications of improved ODI processing for VR and AR applications.

(note: negative points have been addressed in rebuttal with additions to paper)

(JV9Q)
- authors achieved better performance for ODI SR task through 1) a discrete pixel sampling strategy (DPS) and 2) representation considering spherical coordinates.
- coordinate-grid based SR method, LTE is well-known but this paper effectively introduced it to the ERP rescaling task.
- DPS considering geometric properties demonstrates potential to be widely applied across ERP image task. Specifically, DPS was designed for effective learning within ERP datasets, taking viewport rendering into account.
- method has shown significant performance improvements in experimental results.
- Seems valuable to introduce this model to the low-level vision task for ODI.
- paper also demonstrates that SSR can effectively remove the artifacts from ERP images.

(kv6y)
- rescaling and viewport rendering in single pipeline ensures optimization is directly aligned with final visual output on head-mounted displays (HMDs)
- pixel sampling addresses the irregular correspondence issues inherent in spherical images, enabling more accurate and high-quality reconstructions.
- spherical pixel shape representation technique gives better geometric orientation and curvature information, particularly improving visual quality in regions with high latitude and longitude variations, which are typically problematic in conventional methods
- proposed method achieves state-of-the-art performance in multiple viewport rendering tasks. Validation across different fields of view, resolutions, and view directions highlights robustness and versatility of proposed framework.

As a result, I propose to accept this paper as oral presentation. It could even be a candidate for best paper, with the final score of 5.75 after rebuttal.
In case of final acceptance of the paper, authors are strongly encouraged to make sure that all points mentioned in the rebuttal and further reviewer comments are considered for the camera-ready version of the paper.